

# On the link between precipitation and the ice water path over tropical and mid-latitude regimes as derived from satellite observations

Yaniv Tubul[1], Ilan Koren[1], Orit Altaratz[1], Reuven H. Heiblum[1]

[1] Department of Earth and Planetary Sciences, Weizmann Institute of Science, Rehovot, Israel

*Correspondence to*: Ilan Koren (ilan.koren@weizmann.ac.il)

**Abstract.** In this study, we explored the link between clouds' integrated water content and surface rain rate (RR), focusing on deep convective clouds with iced tops. We used a 3-year (2006–2008) global dataset of cloud properties and precipitation rates retrieved from the MODerate-Resolution Imaging Spectroradiometer (MODIS) on Aqua and the Tropical Rain Measuring Mission (TRMM) Multisatellite Precipitation Analysis (TMPA). Previous studies focusing on marine stratocumulus clouds showed a robust link between liquid water path and drizzle rates, in the form of a power law. Consistent with this, we show a quantified link between ice water path (IWP) and surface RR. To reduce the problem to one non-dimensional variable (the power law exponent, β), the IWP and RR were normalized by their local means. We examined the geographical variability of β, and found its mean value to be 1.03 ± 0.13 (1.04 ± 0.12) in the tropics and 1.19 ± 0.19 (1.26 ± 0.20) over the mid-latitudes, during June–August (December–February). The results over the tropical belt showed the best correlation ($R^2 > 0.9$) and lowest standard deviation values, thus the estimations of RR based on IWP measurements for this area are expected to be the most reliable. Such a method offers an estimation of RR using IWP information measured by passive polar-orbiting sensors (such as MODIS). Moreover, it can aid in parameterizing rain properties in regional and global climate models. To enable use of this method, we provide global maps (for June–August) of the required parameters to calculate RR using IWP data.



## 1. Introduction

Clouds and precipitation are key components in the climate system that play an important role in the energy and hydrological cycles (Hartmann et al., 1992; Trenberth et al. 2011; Stephens et al. 2012). Rain is the outcome of many coupled microphysical and dynamic processes within a cloud. These processes span a wide range of scales (from the submicron scale of cloud condensation nuclei to hundreds of kilometers for a cloud field), making rain a challenging parameter for prediction (Waliser et al., 2009; Stephens et al. 2010; Sherwood et al., 2013). Therefore, finding a link between observed cloud properties and the related rain production is a highly desirable target (e.g., Geoffroy et al., 2008; Kostinski, 2008; Kubar et al., 2009; Snodgrass et al., 2009; Koren and Feingold, 2011; Freud and Rosenfeld, 2012; Khain et al., 2013; Lebsock et al., 2013; Hamada et al., 2015).

In-situ measurement studies of low-level warm clouds (especially marine stratocumulus clouds) have shown quantitative links between cloud properties—both microphysical (e.g., cloud droplet size and number concentration) and macrophysical (e.g., cloud-top height, cloud thickness, liquid water path [LWP])—and the produced rain rates (RRs). In particular, a robust link was found between RR and cloud thickness or LWP (reviewed in Geoffroy et al., 2008; Wood, 2012). Cloud-base rain rate ($RR_{cb}$) was found to increase with cloud thickness (H) as $RR_{cb} \propto H^3$ (Pawlowska and Brenguier, 2003; VanZanten et al., 2005) or with LWP as $RR_{cb} \propto LWP^{1.75}$ (Comstock et al., 2004).

The stochastic nature of rain processes implies that clouds with similar properties may yield different precipitation rates and total amounts. Therefore, a valid link between cloud properties and rain yield should be based on a large statistical sample and should be viewed in a climatological manner (i.e., the prediction is likely to represent the average RR for a collection of clouds with given properties). While in-situ measurements can provide direct and accurate high-resolution local data, satellites provide wider spatial and temporal coverage that allows much larger statistical analyses. Satellite measurements are also efficient at covering oceans and remote regions where ground-based measurements are scarce.





In the last few decades, several approaches have been developed for rain estimations based on cloud data derived from passive remote sensing. One method was based on translating infrared (IR) spectral information to rain. This began in the early 1970s, when precipitation rate estimations were carried out by retrieving cloud-top properties using a single thermal IR channel (~11 μm) (reviewed in Arkin and Ardanuy, 1989; Behrangi et al., 2012). The method was based on the assumption that deeper clouds with colder top temperatures are associated with larger RRs. However, such links work better for tropical deep convective clouds, whereas they underperform for cirrus clouds (Arkin and Meisner, 1987) and stratiform precipitation regions in the mid-latitudes. To improve the accuracy of these estimations, the thermal IR channel was combined with passive microwave information, which is more sensitive to precipitation-size drops. This algorithm thus benefited from both techniques' strengths (Adler et al., 1993; Kidd et al., 2003).

Another approach to rain estimation used the retrieved macrophysical and microphysical cloud properties (based on theoretical and empirical links). Some satellite-based studies presented a multispectral approach that uses the visible and near IR (VNIR) channels to retrieve cloud optical thickness (COT), effective radius of cloud-top particles, and cloud water path for various cloud types. Several studies (e.g., Ba and Gruber, 2001; Thies et al., 2008) used this information mainly to separate deep convective clouds from the nearby anvils, or to detect rain areas within stratiform clouds characterized by a homogeneous cloud top. However, their RR estimations were based solely on the IR-retrieved cloud-top temperature. Other studies (e.g. Yan and Yang, 2007; Behrangi et al., 2009; Kubar et al., 2009; Roebeling and Holleman, 2009; Kühnlein et al., 2010; Chen et al., 2011; You and Liu, 2012; Liu et al., 2014) did analyze the sensitivity of the near-surface rain to these (remotely sensed) cloud properties, using primarily original reflectance from the relevant channels (0.56–0.71 μm for COT and 1.5–1.78 μm for cloud-top effective radius), and only a few (Kubar et al., 2009; Roebeling and Holleman, 2009; Chen et al., 2011) used explicit retrieval of these cloud properties.

A new era in measuring rain from space began in the late 1990s with the launching of the Tropical Rain Measuring Mission (TRMM, Kummerow et al., 1998) and CloudSat (Stephens et al., 2002) satellites. In addition to a passive microwave imager, these





satellites carried an active profiling radar that enabled a three-dimensional view of rain systems.

Kubar et al. (2009) used CloudSat and MODerate-resolution Imaging Spectroradiometer (MODIS) to observe the dependence of drizzle intensity on cloud thickness and LWP of shallow marine clouds over tropics/subtropics in the Pacific Ocean. Chen et al. (2011) focused on the same clouds but over the entire global ocean, finding that the LWP has the highest potential for detection and estimation of warm rain. You and Liu (2012) used the TRMM satellite to observe deeper mixed-phase clouds over the entire globe (land and ocean), showing high correlations between the total cloud water path (both liquid and ice) and surface RR, especially along the oceanic tropical belt. However, none of these studies gave any quantification of the links between deep convective cloud properties and rain intensity.

In this work, we describe a robust quantitative link between cloud ice water path (IWP) (retrieved from MODIS) and surface rain (as estimated by TRMM). We focus on deep clouds with ice tops that make a significant contribution to the tropical and mid-latitudinal precipitation budget (Chen et al., 2011).

## 2. Data and Methods

We used cloud properties retrieved by the MODIS algorithm (Platnick et al., 2003) measured on board the Aqua satellite and rain data from the TRMM Multi-satellite Precipitation Analysis (TMPA) product 3B42 V7 (Huffman et al., 2007). The two datasets provide daily data, covering the tropics, subtropics and mid-latitudes from 50°S to 50°N. We focused on June–July–August (JJA) and December–January–February (DJF) in 3 years (2006–2008) using level 3, 1°-resolution daytime data collected around 13:30 local time.

The MODIS algorithm uses the VNIR channels to retrieve the COT and droplet effective radius ($r_{eff}$) (Nakajima and King, 1990). The LWP and IWP are estimated as the product of the COT and $r_{eff}$ (Platnick et al., 2003). The TRMM-TMPA 3B42 product is composed





of precipitation measurements from various microwave instruments, geostationary IR measurements, and surface rain gauges (Huffman et al., 2007). This product is available from 1998, at 3-h and 0.25° resolution.

To obtain simultaneous cloud and rain measurements, we developed an Aqua time
equivalent precipitation database by sampling the measured rain data that were closest to the Aqua's passing time. For each day and location, the Aqua passing time (~13:30 local time) was translated to UTC time and the weighted average of two TRMM measurements around it were used to estimate the local RRs (Koren et al., 2012).

The LWP or IWP are expected to correlate with rain amounts as they represent the
condensate mass of the cloud, which is the source of rain. In the case of warm clouds, it has been shown to obey a power law relation (Comstock et al., 2004). The TRMM-TMPA is tuned to estimate surface RRs that are stronger than 0.7 mm h$^{-1}$, which generally characterize relatively deep clouds. Indeed, most of the pixels that contained MODIS cloud data and detectable rain by TRMM over our study region (50°S to 50°N)
had ice tops (70% of the pixels with available data for JJA 2007). Sensitivity tests have shown that over the study area, IWP is the cloud-retrieved variable that correlates best with the surface rain. Therefore, in this work, we chose to focus on ice-top clouds and use the MODIS IWP as the main cloud parameter.

Estimation of the link between IWP and RR requires a compromise between the size of
the averaged dataset and the locality of the results. On the one hand, more pixels yield better statistics, which is especially important for rain data that are characterized by high variance. On the other, more pixels cover a larger area and therefore the results are less localized and more likely to average different meteorological conditions. We tested averaging scales between 3 and 11 pixels (with 1° resolution). All averages showed
similar trends and here we chose to present the 5 x 5 pixel analyses which were shown to be a good compromise between statistics and locality. All of the analyses presented here were therefore done with a moving 5 x 5 pixel window (in 1-pixel steps).

RRs were shown to be linked to clouds' LWP (in our case IWP) by the mean of a power law (Comstock et al., 2004)





$$RR = \alpha \times IWP^{\beta} \tag{1}$$

where $\beta$ is the power law exponent and $\alpha$ is a scaling factor. Here, we reduce the problem to one non-dimensional variable by normalizing the data by its local mean in the following way: for each N x N (in our case N = 5) window, we calculated the RR and IWP averages and replaced the variables by their normalized values: $\Gamma = \frac{RR}{\overline{RR}}$ and $\Lambda = \frac{IWP}{\overline{IWP}}$.

Using Eq. (1):

$$\Gamma = \frac{RR}{\overline{RR}} = \frac{\alpha \times IWP^{\beta}}{\overline{\alpha \times IWP^{\beta}}} \tag{2}$$

Multiplying Eq. (2) by $\left(\frac{1/\overline{IWP}}{1/\overline{IWP}}\right)^{\beta}$ yields

$$\Gamma = \frac{\Lambda^{\beta}}{\overline{\Lambda^{\beta}}} = \Theta \Lambda^{\beta} \tag{3}$$

We note that per window, $\Theta = \frac{1}{\overline{\Lambda^{\beta}}}$ is a constant that can be calculated using the cloud data only once $\beta$ is given. Taking the logarithm of both sides of Eq. (3) linearizes the problem so that the exponent $\beta$ can be easily estimated from the data:

$$\log \Gamma = \log \Theta + \beta \times \log \Lambda \tag{4}$$

To analyze the slope $\beta$ (based on Eq. (4)), RR and IWP data points located within each 5 x 5 window were first normalized to $\Lambda$ and $\Gamma$ by their means and then $\beta$ was estimated as the best linear fit between their logarithms.

## 3. Results

As a demonstration of our methodology, Fig. 1 presents an example of the association between IWP and RR for a single 5 x 5 box located over the tropical Atlantic (see the marked square in Fig. 1a). The figure presents maps of the mean IWP ($\overline{IWP}$, g m$^{-2}$, Fig. 1a) and mean RR ($\overline{RR}$, mm h$^{-1}$, Fig. 1b) for JJA 2007 (92 days). Figure 1c presents a scatter plot of the dependence between the logarithms of the normalized variables. The





data used for the scatter plot are divided into 50 bins that contain equal numbers of samples.

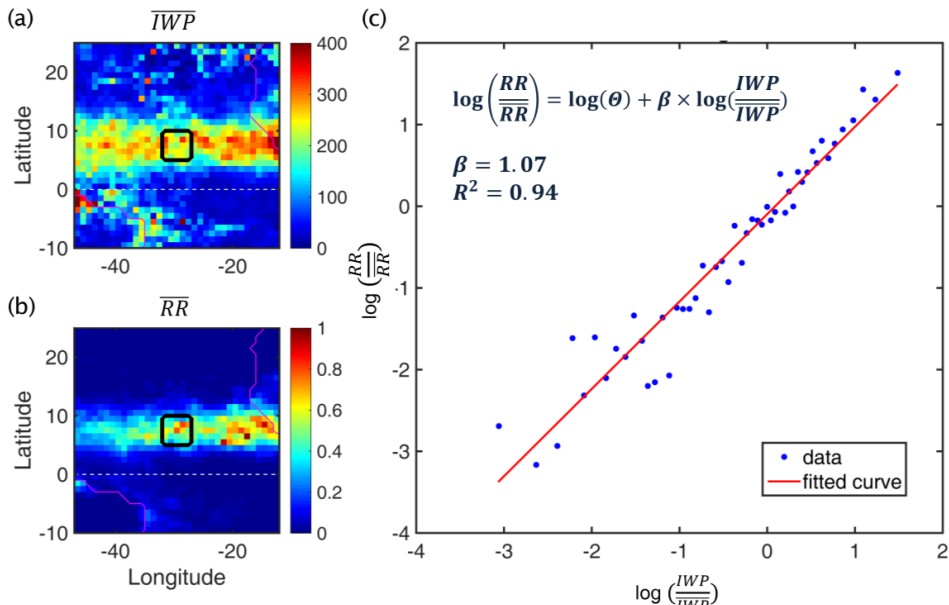

**Figure 1: Mean maps of (a) ice water path (IWP, g m⁻²) derived from MODIS-Aqua and (b) surface rain rate (RR, mm h⁻¹) derived from TRMM-TMPA for June–July–August 2007. (c) Scatter plot of $\log(\frac{RR}{\overline{RR}})$ against $\log(\frac{IWP}{\overline{IWP}})$ using daily data for the entire season (92 days) and for a region of 5° × 5° (black square in (a)). The data are divided into 50 bins, each with 22 samples. The red line represents the linear fit with slope $\beta = 1.07$ and $R^2 = 0.94$.**

Similar to what was done in Fig. 1, we examined the linear correlations between RR and IWP over the tropics, subtropics, and mid-latitudes, extending between 50°S and 50°N. The average IWP and RR for JJA 2007 are presented in Figs. 2a,b, and the correlation ($R^2$) and β slope values are presented in Fig. 2c,d. More than 60% of the pixels with available RR and IWP information were characterized by correlation values above 0.8 (Fig. 2c, marked in yellow and red colors). These pixels included the entire tropical belt and most of the mid-latitudes (as also shown in Fig. 3a). The mean β value for the entire study area (where $R^2>0.8$) was 1.08 with a standard deviation of 0.18, and relatively




higher values seen in the mid-latitudes. Areas with insufficient data for analysis (colored in black, Figs. 2c,d) were located mostly over the subtropics. These areas are characterized by shallow clouds that produce weak rain, or by clear skies.

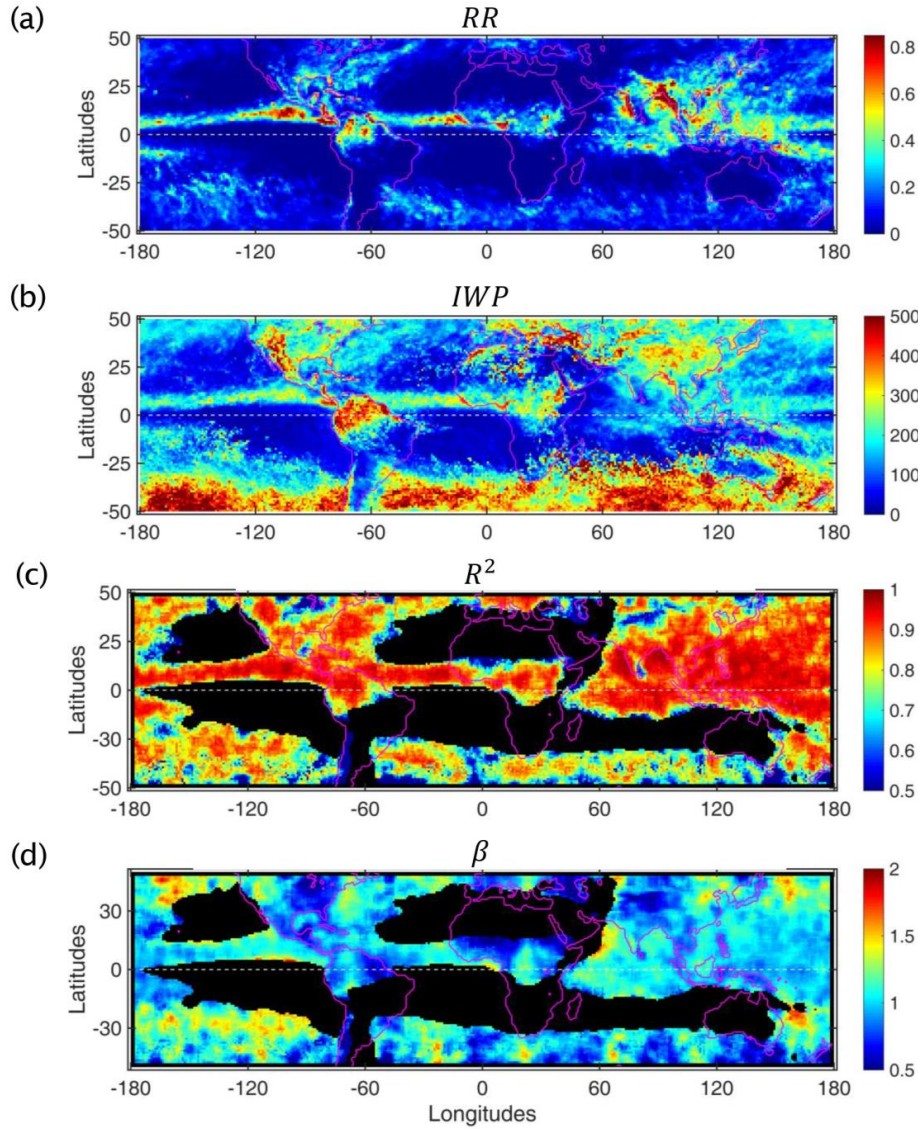

Figure 2: Near-global distribution maps of (a) rain rate (RR, mm h$^{-1}$), (b) ice water path (IWP, g m$^{-2}$), (c) R$^2$ and (d) β exponent (i.e., slope) for June–July–August 2007. The β value was estimated based on Eq. (3). The resolution of the R$^2$ and β value maps is 1° but calculations for each pixel were based on a box of 5 x 5 pixels surrounding that pixel.





To examine the geographical dependence of the β distributions in more detail and to compare the tropical belt to higher latitudes, we selected four regions with mostly $R^2 > 0.8$ pixels. Each histogram in Fig. 3b shows the distribution and mean of β values within the corresponding 10 x 20 pixel boxes in Fig. 3a. Regions located outside the tropical belt

(boxes 3 and 4) were characterized by larger β values and a larger variance. Figure 3c shows two histograms of β values, one for the entire tropical belt (20° S – 30° N) and the other for the mid-latitude regimes (20° – 50° S and 30° – 50° N). The tropical region that exhibited the higher $R^2$ values (see Fig. 2c) had a mean β value of 1.01 with standard deviation of 0.13, whereas the mean β value of the mid-latitudes (with $R^2 > 0.8$) was 1.18

with standard deviation of 0.20 (see Fig. 3c).

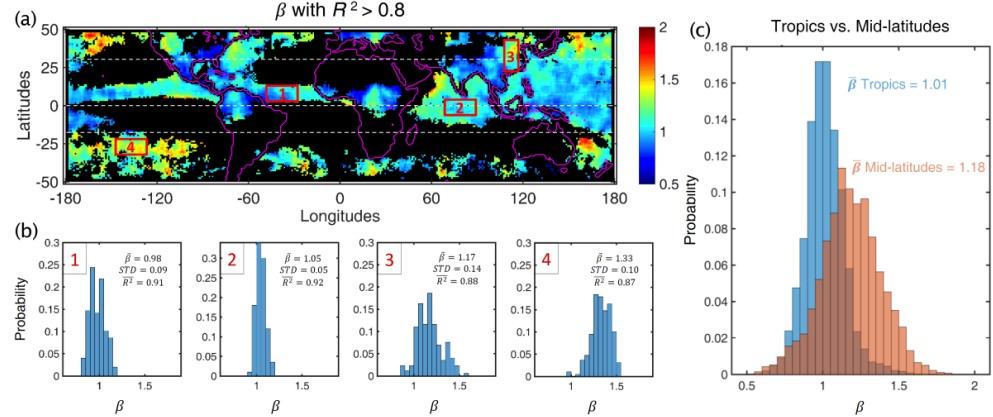

**Figure 3: (a) Near-global distribution map of β values characterized by $R^2 > 0.8$ during June–July–August 2007. (b) Histograms (1–4) of the β values over four selected boxes (10 x 20 pixels), correspondingly numbered in (a), with the local mean β value, standard deviation and mean $R^2$ (see text in each histogram). (c) β histograms for**

**the tropics (20° S – 30° N, blue) and the mid-latitude regions in both hemispheres (20° – 50° S and 30° – 50° N, red).**

To examine the generality of our results, we expanded the analysis to include multiple years and two different seasons. Figure 4 shows maps of β values and their corresponding

histograms for 3 years (2006–2008) and 2 seasons (JJA compared to DJF). The mean exponent value for the entire domain was 1.09 ± 0.18 (1.12 ± 0.19) for JJA (DJF) in 2006–2008. The results shown for JJA 2007 (Figs. 2, 3) were consistent with the other



years and season. The tropics were characterized by a relatively lower β value and lower standard deviation (1.03 ± 0.13 in JJA and 1.04 ± 0.12 in DJF) compared to the mid-latitudinal belts (1.19 ± 0.19 in JJA and 1.26 ± 0.20 in DJF). The borders between the tropics and mid-latitudes were positioned differently for each season due to seasonal

5 migration of the Hadley Cells. Thus the tropics spanned latitudes 20° S – 30° N (25° S – 25° N) and accordingly, the mid-latitude regimes spanned 20° – 50° S and 30° – 50° N (25° – 50° S/N) in JJA (DJF). All analyses showed smaller mean β and standard deviation values of the tropics compared to the mid-latitudes.

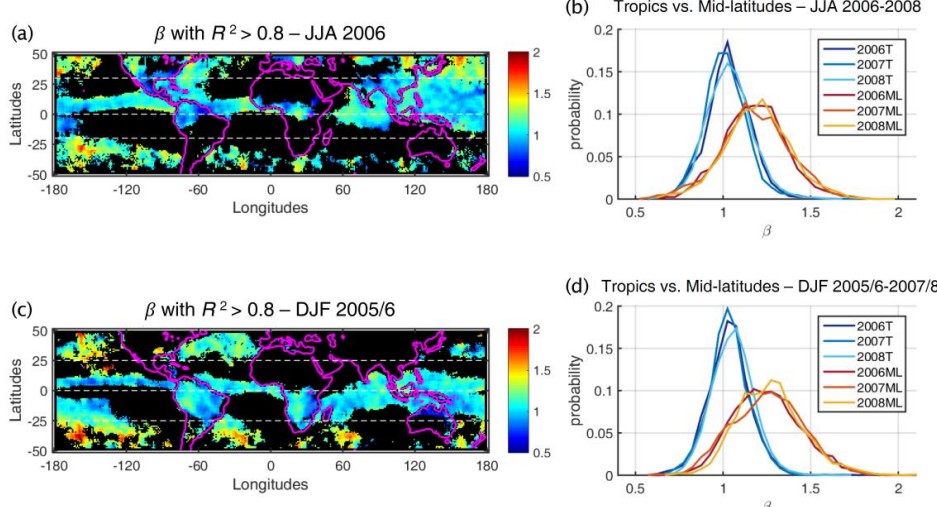

**Figure 4: (a) Near-global distribution map of β values characterized by $R^2 > 0.8$ during June–July–August (JJA) 2006. (b) β-value histograms for JJA in 3 different years (2006–2008), divided for the tropics (20° S – 30° N) and mid-latitude regimes of both hemispheres (20° – 50° S and 30° – 50° N). (c, d) Similar to (a, b) but for the months December–January–February (DJF) 2005/6 (c) and DJF 2005/6-2007/8 (d). Dashed lines in each map mark the**

15 **equator along 0° latitude and the tropics/mid-latitude borders.**

We note that $R^2$ is a measure of how closely the linear slope calculations (in the $\log(\frac{RR}{\overline{RR}})$ vs. $\log\left(\frac{IWP}{\overline{IWP}}\right)$ space) reflect the data in each scatter plot, whereas the standard deviations

20 reflect the variance in slopes for a given area. Therefore, a larger standard deviation does not necessarily reflect a reduction in the quality of the slope calculations. It may simply



reflect a larger natural variance in mid-latitudes compared to the tropics. A possible source for this larger variance might be the type of meteorological conditions in the mid-latitudes that produce such RRs. Unlike the tropics that are controlled mostly by local convection, the mid-latitudes' convective rains are more often associated with frontal
systems. Those systems experience rather sharp discontinuities in thermodynamic conditions which may result in large variance when translating cloud properties to rain.

## 4. Summary and Discussion

Previous studies of boundary-layer precipitating clouds have shown a robust link between cloud properties (such as water path and thickness) and drizzle rates. In this work, we expand this view to deeper clouds from the tropics to the mid-latitudes. To do so, we merged cloud information from MODIS with RR data based on TRMM and combined with other satellites. The TRMM product is sensitive to the relatively strong RRs (>0.7
mm h$^{-1}$) that are likely to be produced by deeper clouds with ice tops. Our initial sensitivity test showed that the best correlation of the TRMM-based RR is with the MODIS IWP. Therefore, for cloud properties, we merged the MODIS IWP with the TRMM-TMPA RR by sampling the rain data near Aqua overpass times. Based on previous studies, we looked for a power law link ($RR \propto IWP^{\beta}$) between the variables. To
get a non-dimensional exponent, we normalized the RRs and IWPs by their local means and estimated the linear slopes between $\log(IWP/\overline{IWP})$ and $\log(RR/\overline{RR})$ for 5 x 5 pixel boxes (each pixel being 1°).

Our results show that the slope values are well confined. Nearly 65% of the pixels
containing ice-topped clouds and precipitation showed correlations higher than 0.8 between these two parameters. The dependence of RR on IWP was shown to be nearly linear, with a mean exponent value (β) of 1.09 ± 0.18 (1.12 ± 0.19) for the entire domain for JJA (DJF) in 2006–2008 (Fig. 4). The tropics were characterized by smaller mean β value and standard deviation (Fig. 4). These results were consistent for different years
and seasons with 1.03 ± 0.13 (1.04 ± 0.12) for the tropics and 1.19 ± 0.19 (1.26 ± 0.20) for the mid-latitudes during JJA (DJF).



The provided information on the regional mean exponent of the slopes and their standard deviations can help in estimations of rain based on cloud water path information for previous years (with no space-radar data) or for areas where rain measurements are lacking. It can also help improve rain parameterization in regional and global models. To

estimate RR based on IWP according to the method described in this work, we provide (see supporting materials) the 3-year JJA averaged maps of α and β parameters together with the $R^2$ as calculated using the logarithmic form of Eq. (1) ($\log(RR) = \log(\alpha) + \beta \times \log(IWP)$). Future observational and theoretical studies are required to link the spatial and temporal exponent variability to changes in cloud type, cloud microphysical or

macrophysical properties, convective/stratiform precipitation fraction, and rain intensity, so that we can gain a more thorough understanding of the physical mechanisms behind such links.

**Acknowledgements.** The research leading to these results received funding from the European Research Council under the European Union's Seventh Framework Programme (FP7/2007-2013)/ERC Grant agreement no. 306965.

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
