# Peer review of "On the link between precipitation and the ice water path over tropical and mid-latitude regimes as derived from satellite observations"

_Atmospheric Measurement Techniques, 2017_

## Referee Comment (RC1) · Anonymous Referee #1 · 22 May 2017

General Comments: This manuscript relates ice water path (IWP) and rainfall rate to convection in the tropics and midlatitudes, whereas the previous published literature had focused mainly on the IWP-rain rate relationship for shallow stratiform clouds. The paper is well-written and would be of significant interest to readers of Atmospheric Measurement Techniques; however, I have significant reservations about the validity of the results because TMPA 3B42 was used as the ground validation dataset and the internal inconsistencies in the data set (it uses rain rates based on infrared and passive microwave in different places) and because the microwave retrievals may contain implicit relationships between IWP and rain rate that may compromise the significance of the results.

[Figure]

Specific Comments: 1. Page 4, lines 20-21: The use of TMPA 3B42 as the rainfall "ground truth" is a significant concern. 3B42 consists of rain rates retrieved from passive microwave (PMW) instruments and (where they are not available) rainfall rates from infrared (IR) brightness temperatures calibrated against PMW. Because of the relatively infrequent sampling of PMW in many locations, the IR rain rates will have a significant influence on 3B42 and IR rain rates have well-documented shortcomings because they rely on the relationship between cloud-top brightness temperature and surface rainfall rate (which is at least much more robust for convective clouds than for stratiform clouds). Furthermore, the PMW portion of the 3B42 fields is based on sensitivity to ice hydrometeors (or liquid if present in sufficient numbers) and hence some of the relationships between IWP and rain rate observed may be an artifact of the retrieval and not necessarily real. Although it would significantly reduce the amount of available validation data, using a consistent dataset such as the TRMM PR alone (which, because of its inclined orbit would occasionally intersect the Aqua orbital path) would reduce the effects of these retrieval artifacts and significantly increase confidence in the manuscript's findings 2. Page 4, line 22: 3B42 is actually 3-hourly time resolution; this is stated in lines 2-3 of page 5 but should probably be stated here instead for clarity. It might be even better to reorganize these first two paragraphs of Section 2 to completely describe the MODIS cloud products in one paragraph and 3B42 in the next. 3. Page 4, line 20: The MODIS cloud property retrievals are at the pixel level (1-km resolution) and 3B42 is at 0.25° lat / lon resolution, so what was the reason for aggregating up to 1° lat / lon resolution? 4. Page 5, line 5: These rain rates should be described as "retrieved" or "estimated", not "measured" since strictly speaking only a gauge provides a direct measurement. 5. Page 5, lines7-8: The 3B42 rainfall rates represent an average over the entire 3-hour period rather than a value at a particular point in time; therefore, interpolation is probably not recommended; rather, the Aqua overpass should be matched with whichever 3-hour window of 3B42 is coincident with it. 6. Page 9, line 9: 3B42 provides estimates of rain rate, not rain amount. 7. Page 5, lines 19-27: It might be better to introduce this information after describing the calculation of the local means on line 5 of page 6 so that the purpose of the averaging is understood. 8. Page 5, lines 25-26: What is meant by the "trends" in line 25? Also, what was the basis determining what was "a good compromise between statistics [statistical significance?] and locality?" 9. Page 6, Equation (3): It would appear that when simplifying Eq. 2, the $\alpha$ terms drop out and all that is left is $\Lambda\beta$. Where does $\theta$ come from here (and, consequently, in Eq. 4)? 10. Page 7, lines 1-2: why were the data divided into 50 bins prior to creating the scatterplot, and what was the basis for binning? At first glance it would appear that this would make the results appear much less noisy than they really are. 11. Page 9, line 8: this relationship is nearly linear ($\beta$=1); it would be informative to test for the statistical significance of $\beta$<>1. 12. Page 9, lines 7-10 and elsewhere: it might be helpful to provide a plot or a few values showing how different the RR-IWP relationships are for typical ranges of IWP so that the significance of these differences (or similarities) between regions is clearer.

Technical Corrections: 1. Page 2, line 3: Trenberth appears to be the sole author of the paper cited here. 2. Page 3, lines 17-18: There is no entry for Thies et al. (2008) in the References. 3. Page 13, lines 23-25: Hou et al. (2014) is not cited in the body of the manuscript. 4. Page 14, lines 22-27: Lebsock et al. (2011) and Lin and Rossow (1997) are not cited in the body of the manuscript.

---

## Referee Comment (RC2) · Anonymous Referee #2 · 13 Jun 2017

This manuscript reports on correlations between gridded precipitation and ice water path observations from TRMM and MODIS, respectively. The paper seeks to add to the literature concerning algorithms for retrieving convective precipitation intensity from visible and infrared radiance measurements through a somewhat different pathway that first involves retrieving IWP as opposed to directly correlating radiances to surface rainfall rate. The subject is appropriate for Atmospheric Measurement Techniques but there are fundamental with the manuscript that make it unsuitable for publication at this time. These include critical deficiencies in the approach, the omission of any direct evaluation of the technique, and a failure to acknowledge some fundamental circularity in the logic. Specifically, no justification is offered concerning the choice of datasets

used in the study, there is no discussion of the impact of uncertainties in the IWP and RR estimates including the representativeness of visible and infrared-derive IWP in convective storms, and the physical significance of correlating rainfall and IWP at a resolution of 5x5 (i.e. much larger than the scale over which precipitation processes occur) is not discussed. In addition, little evidence is provided to support the claim that this technique might offer a more robust method for retrieving rainfall rate than those that have been developed in the past. Indeed, the very dataset the authors adopt to develop their statistical regressions already incorporates infrared estimates of rainfall intensity information that they purport to replace with this new algorithm. As a result of these concerns, which are elaborated below, I cannot recommend publication of this paper at this time.

Specific Comments:

1. Despite the authors claims concerning the novel nature of this technique, it is not clear how this work advances the state of rainfall intensity retrievals from visible and infrared radiances that already exist in the literature. While it is clear that ice water and rainfall intensity are physically connected through ice-phase precipitation processes, it is not clear that the specific approach presented here is any different from simply using these radiances directly to infer rainfall rate. By regressing separate IWP retrievals against rainfall rate, the method simply introduces the additional step of first estimating IWP from the raw radiances that can introduce its own uncertainties including assumed particle size, shape, vertical structure, sensitivity to large particles, and saturation at high IWP that complicate the subsequent relationship to rainfall rate. There is no mention of the influence of these sources of uncertainty in the manuscript. In fact, based on the fact that the authors use existing cloud products and do not perform any retrievals themselves, it is not clear they are aware of these issues or the fact that the measure of IWP used here is likely far from optimal for precipitating cloud scenes.

2. More problematic is the fact that the precipitation dataset used in this analysis actually includes geostationary infrared radiance information in it. This guarantees a

relationship between the TRMM precipitation estimates and associated cloud fields since similar observations influence both retrievals. Other than a quick mention of this at the top of page 5, this circularity is not discussed at any length in the manuscript.

3. Another concern centers on the lack of error bars on any of the component datasets used in the analysis. What is the accuracy of the rainfall rates and ice water path estimates used? How does the accuracy of these products vary with rainfall intensity? The IWP estimates almost certainly 'saturate' in more convective cloud regimes and visible and infrared measurements are generally not sensitive to precipitation-sized particles from which rain actually forms. No attempt is made to compare the MODIS IWP estimates to those from collocated passive microwave observations provided by TRMM.

4. While the sensitivity of the results to averaging-scale was 'tested' according to the authors, it is still not clear how the physical processes described earlier in the manuscript relate to IWP and rainfall rate estimates over a 5x5 grid box. How much of the 'signal' emerging from the regressions shown in Figure 1 is simply caused by the correlation between cloud fraction and rain fraction within these large boxes? Has any effort been made to normalize the results by cloud fraction and rain fraction to remove such effects? I do not believe that normalizing by the mean IWP and R over each grid box fully accomplishes this.

5. Each of these concerns brings up one additional overarching concern related to the choice of datasets adopted for this analysis: it is not at all obvious that the datasets adopted here are optimal for this study. Since TRMM carries the Visible and InfraRed Scanning radiometer (VIRS), ice water path estimates similar to those derived from MODIS observations can be compared against TRMM rainfall estimates from either the Precipitation Radar (PR) or Microwave Imager (TMI) directly avoiding the issues of collocating MODIS and TMPA. Collocating these two datasets from independent satellites merely introduces uncertainties owing to time/space collocation errors, restricts the time of day to 1:30 pm, introduces artificial correlations through the influence of IR

measurements on the TMPA, requires analyzing larger grid boxes (see below), and precludes the use of passive microwave IWP estimates for independent validation. What's more, many of the visible/infrared rainfall retrieval techniques currently employed have been developed using these direct collocated datasets and are, therefore, likely more robust than the technique proposed here.

6. That leads me to my final point – no independent evaluation or comparisons to other techniques is provided to support the claims concerning the benefits of this new approach. Evaluation against independent rainfall estimates is required to assuage fears concerning the flaws outlined above.

---

## Author Comment (AC1) · 12 Oct 2017

The comment was uploaded in the form of a supplement:
https://www.atmos-meas-tech-discuss.net/amt-2017-121/amt-2017-121-AC1-supplement.pdf

---

## Author Comment (AC2) · 12 Oct 2017

**Final Author Comment – Tubul et al., AMT 2017**

**Reply to the reviewers**

We would like to thank the reviewers for their efforts and comments that helped us clarify this manuscript and make it more accurate. Before answering all the comments (in a point by point manner) we would like to open with a clarification about the motivation and main objective of this study. Our motivation was to investigate the link between clouds' integrated water content and surface rain rate (a power-law) using satellite-based observations. The focus of this research and its suggested application is for large spatial and temporal scales. Therefore we choose to analyze seasonal data sets, in daily resolution and 1° spatial resolution. For this mission we used two of the most tested and validated quasi-global gridded products of cloud properties and rain rate that are optimal for our current study; the MODIS-Aqua and the TRMM-TMPA. Both products span 15 - 20 years of measured data, together with continues research, evaluation and on-going development and improvement. As suggested in numerus studies, observations and parameterization of the link between surface rain rate (RR) and cloud properties are required for better understanding of the physical processes controlling rain formation and for improving cloud and rain description in climate models. Our suggested method is applicable to past and present passive optical observations of cloud properties such as the Advanced Very High Resolution Radiometer (AVHRR), and present to future missions such as the Suomi National Polar-Orbiting Partnership (Suomi NPP) that is carrying the Visible Infrared Imaging Radiometer Suite (VIIRS, Cao et al., 2013), and MODIS. These instruments' measurements enable estimations of surface RR based on cloud properties in global and seasonal to climatological scales.

Both reviewers were concerned about the circularity in the logic of our method. We would like to highlight the fact that the TRMM-TMPA dataset that we are using as our surface rain measure, although partially relying on passive microwave (PMW) measurements that are sensitive to ice water path (Huffman et al., 2007; Behrangi et al.,

2012), it is calibrated by the TRMM's TMI-PR product and by surface rain gauges. In the current TRMM-TMPA version (V7) for example, brightness temperature detected by the main PMW-based satellites (e.g. TMI, AMSR-E, SSMI, and SSMIS) is translated to surface RR based on collection of real observations form the active PR, and not by cloud model computational data as in previous versions (Huffman and Bolvin, 2017). Such calibrations together with evaluation of the product by surface rain radars support a more direct association of these estimations with rain-size hydrometeors and surface rain measurements.

We respond to all reviewers' comments and conducted additional analyses to better support this work. The main revisions in the manuscript:

1. We show the relatively small fraction of the IR measurements within our rain dataset and their limited coverage, concluding it is not a significant component in our analysis.

2. We estimate the uncertainty in the exponent value and present the statistical significance of the linear exponent over the tropical belt.

3. We evaluate our results using independent rain product which is the PERSIANN-CCS that does not rely primarily on passive microwave (PMW) products but on IR rain estimations and machine learning algorithm. The high similarities of the power law exponents ($\beta$) calculated using this product are presented.

We have addressed all of the reviewers' comments below point by point. Each of the reviewers' question or remark is followed by our answer (written in blue).

**Referee #1**

**General Comment:** This manuscript relates ice water path (IWP) and rainfall rate to convection in the tropics and mid-latitudes, whereas the previous published literature had focused mainly on the IWP-rain rate relationship for shallow stratiform clouds. The paper is well-written and would be of significant interest to readers of Atmospheric Measurement Techniques; however, I have significant reservations about the validity of the results because TMPA 3B42 was used as the ground validation dataset and the internal inconsistencies in the data set (it uses rain rates based on infrared and passive microwave in different places) and because the microwave retrievals may contain implicit relationships between IWP and rain rate that may compromise the significance of the results.

**Answer**: Thank you for the effort and important comments. We answer each of the specific comments below, trying to clarify issues and solve the raised problems. Find our detailed answers (written in blue) following each of your specific comments.

**Specific Comments:**
1. Page 4, lines 20-21: The use of TMPA 3B42 as the rainfall "ground truth" is a significant concern. 3B42 consists of rain rates retrieved from passive microwave (PMW) instruments and (where they are not available) rainfall rates from infrared (IR) brightness temperatures calibrated against PMW. Because of the relatively infrequent sampling of PMW in many locations, the IR rain rates will have a significant influence on 3B42 and IR rain rates have well-documented shortcomings because they rely on the relationship between cloud-top brightness temperature and surface rainfall rate (which is at least much more robust for convective clouds than for stratiform clouds). Furthermore, the PMW portion of the 3B42 fields is based on sensitivity to ice hydrometeors (or liquid if present in sufficient numbers) and hence some of the relationships between IWP and rain rate observed may be an artifact of the retrieval and not necessarily real. Although it would significantly reduce the amount of available validation data, using a consistent dataset such as the TRMM PR alone (which, because of its inclined orbit would occasionally

intersect the Aqua orbital path) would reduce the effects of these retrieval artifacts and significantly increase confidence in the manuscript's findings.

**Answer:** We thank the reviewer for this important comment. We used the TRMM-TMPA dataset for rainfall estimations since it is highly tested product that is based on multiple data sources, and it is calibrated with active precipitation radar (PR). These calibrations together with validations of the product against surface radars make it a reliable product for surface rain. Many previous studies evaluated the performance of the TRMM-TMPA 3B42 V7 surface rain and found good agreement with surface measurements (e.g. Chen et al., 2013a, 2013b; Xue et al., 2013; Peng et al., 2014; Tang et al., 2015; Deo et al., 2017), especially for moderate to high rainfall rates. Of course we are aware of its limitations and know it is not rainfall "ground truth", nevertheless it is the best global dataset we can currently use for this purpose.

In order to evaluate the influence of the IR estimations on the precipitation data we used the *satellite precipitation source* provided by the TMPA 3B42 dataset to show the fraction of the IR out of the total data. The source probability histogram (Fig. 1a) shows that the IR estimations (i.e. source number 50, Huffman and Bolvin, 2017) are less than 10% of the total observations used in our dataset of JJA 2007. Moreover, we examined the spatial distribution of the IR measurements fraction (Fig. 1b) and found it is significant only in several spots over the southern hemisphere's mid-latitudes (latitudes 20° to 50°S). Along the tropical belt the IR fraction is also no more than 10% of the data (not shown). Note that this analysis was done using single 3-hr time frames (without averaging) in order to enable tracking of the observations source.

[Figure]

Figure 1: (a) Probability histogram of satellite precipitation source provided by the

TRMM-TMPA 3B42 V7 in the Aqua time database for June–July–August 2007 (JJA 2007). Source index #50 is for the IR estimations. The other sources within our dataset are the AMSU (index #1), TMI (#2), AMSR (#3), MHS (#6), AMSU&MHS avg. (#30), and conical estimates avg. (i.e. TMI, AMSR-E, SSMI, and SSMIS, #31) (b) Near-global distribution map of IR estimations fraction out of the total number of estimations for each pixel in the same database as in (a).

In addition, we calculated the power law exponent using only the High Quality (HQ) precipitation estimations. The HQ product combined all the PMW precipitation estimations available from TCI, TMI, SSMI, SSMIS, AMSR-E, AMSU-B, and MHS. We compared the exponents calculated using the HQ precipitation with exponents calculated using all observational sources (i.e. PMW+IR, the precipitation variable) as in the paper (Fig. 2). The two exponent maps (Fig. 2 a-b) show a very little difference in the exponent values and distribution except for areas where the IR data has larger contribution. In areas such as the southern mid-latitudes the main differences are the spatial coverage of the power law exponent (i.e. $\beta$) and small differences in the $\beta$ values. $R^2$ histograms on the right-top (Fig. 2 c-d) for the tropics and mid-latitudes areas strengthen our findings by showing very small difference in the $R^2$ distribution. For the $\beta$ value histograms (Fig. 2 e-f) we find almost no difference in the tropics but small decrease in the $\beta$ values of the product that includes the IR in the mid-latitudes. This is due to larger contribution of the IR estimations in the southern hemisphere, with relatively smaller $\beta$ values than for the PMW only product. Such findings demonstrate the small and limited influence of the IR estimations on $\beta$.

[Figure]

Figure 2: (a) Near-global distribution map of $\beta$ value characterized by $R^2 > 0.8$ using the TRMM-TMPA 3B42 precipitation product (PMW+IR) during June–July–August (JJA) 2007. (b) Similar to (a) but using the TRMM-TMPA 3B42 HQ precipitation product (PMW only). (c, e) probability histograms of $R^2$ and β-value for both precipitation products during JJA and for the tropics (20°S – 30°N). (d, f) similar to (c, e) but for mid-latitudes regimes of both hemispheres (20° – 50°S and 30° – 50°N)

One fundamental advantage of the TRMM HQ product over other PMW rain products is the calibration of the PMW measurements with the TRMM active PR. All the PMW estimations used by the TMPA 3B42 product were inter-calibrated to the TRMM's TMI-PR product (i.e. 2B31). In the current version (V7) the primary PMW-based instruments are the TMI, AMSR-E, SSMI, and SSMIS. For these instruments, observed brightness temperatures were translated to surface RR using an extensive library of hydrometeor profiles relating microwave brightness temperatures to surface precipitation rates based solely on the PR data (Huffman and Bolvin, 2017). In pixels that none of the TRMM's instruments or these HQ estimations are available during the 3-hour window, PMW estimations from the AMSU-B and MHS are used. Rain rate estimations using these last instruments are based on IWP–precipitation rate relations but they are also calibrated

with the TMI-PR product.  These calibrations together with evaluation of the TRMM-TMPA 3B42 product against surface measurements and surface radars make it more reliable rain product.

Finally, regarding the suggestion of the reviewer to use only the PR data in the analysis. Unfortunately this is impossible since it does not provide us the needed dataset. Our main objective is characterization of the link between MODIS cloud properties and rain rate in a seasonal manner and on a large-scale. The product for such mission should be a global gridded product with as large as possible spatial and temporal coverage, large statistics, and multiple sources of rain information. The PR data alone does not support these requirements.  The TRMM-TMPA 3B42 in a daily 1° resolution is suitable product for that (and the MODIS-Aqua L3, as well).

For demonstrating the similarity of our general analysis results to a limited analysis that is based on specific sensors we used the *satellite precipitation source* information. In order to enlarge the statistical sample to enable such type of analysis we added to the PR-TMI estimations and the PMW estimations that were translated to RR using the PR profiles library. This is the minimal selection of sources (three sources) that enables this type of analysis, and we selected the best of them. We calculated the $R^2$ and $\beta$ value using these sources and compared it with the general analysis as performed in the paper. In order to create source-based precipitation dataset we had to change our method of analysis from a weighted average of two 3-hour time frames to a non-averaged single frame. To check the consistency with the original analysis (from the paper) we compared it first with a non-averaged general dataset, including all the observational sources. Just after verifying this similarity we filtered the required data sources; the PR-TMI and PR-calibrated PMW. Presented below (Fig. 3) is a comparison between the four datasets including the precipitation (the original general analysis), HQ-precipitation (including only PR-TMI and PR-calibrated PMW), all precipitation sources with no-average, and finally precipitation with no-average and filtered sources. We show probability histograms of $R^2$ and $\beta$ values, divided for the Tropics and Mid-latitudes, using pixels with $R^2>0.8$. The $R^2$ distributions (Fig. 3 a-b) for all sources with no-averaged is consistent with the original general dataset but the filtered-source shows a decrease in the $R^2$ over the tropics and an increase of $R^2$ over mid-latitudes. In the $\beta$ histograms (Fig. 3 cd) we found a decrease in the mean β value for both no–averaged datasets. The similarity in the trend is due to the dominance of the PMW measurements in both collections over the tropics. On the other hand, over the mid-latitudes the no-averaged dataset has similar β distribution as for the general (precipitation) dataset while the filtered-source collection has β distribution similar to the HQ-precipitation. This trend is due to the contribution of the IR source to both collections including it. In order to have a large enough statistical sample we also increased the analyzed box of the filtered-source from 5 by 5 to 7 by 7.

[Figure]

Figure 3: (a, c) Probability histograms of $R^2$ and β-value for Aqua time databases using different products: precipitation (PMW+IR), HQ-precipitation (PMW only), precipitation

with no average, specific precipitation source (PR, TMI, PR-calibrated) with no average, during JJA and for the tropics (20° S – 30° N). (b, d) similar to (a, c) but for mid-latitude regimes of both hemispheres (20° – 50° S and 30° – 50° N).

We revised the text in several places in the manuscript.

In the Data and Methods section: *"Surface rain rate data was taken from the TRMM Multi-satellite Precipitation Analysis (TMPA) product 3B42 version 7 (Huffman et al., 2007). This product contains precipitation estimations based on various microwave instruments, geostationary IR, and surface rain gauges (Huffman et al., 2007). Fundamental advantage of this product is the inter-calibration of the PMW rain estimations with the TRMM active precipitation radar (PR). IR measurements are used only when PMW observations are not available (in our dataset it is less than 10% of the data, and it is located mainly over the southern mid-latitudes). Many previous studies evaluated the performance of the 3B42 V7 surface rain and found good agreement with surface measurements (e.g. Chen et al., 2013a, 2013b; Xue et al., 2013; Peng et al., 2014; Tang et al., 2015; Deo et al., 2017), especially for moderate to high rainfall rates."*

2. Page 4, line 22: 3B42 is actually 3-hourly time resolution; this is stated in lines 2-3 of page 5 but should probably be stated here instead for clarity. It might be even better to reorganize these first two paragraphs of Section 2 to completely describe the MODIS cloud products in one paragraph and 3B42 in the next.

**Answer**: We thank the reviewer for this comment. We reorganized the first two paragraphs of the Data and Methods section as suggested.

3. Page 4, line 20: The MODIS cloud property retrievals are at the pixel level (1-km resolution) and 3B42 is at 0.25° lat/lon resolution, so what was the reason for aggregating up to 1° lat /lon resolution?

**Answer**: We thank the reviewer for this comment that helped us clarify this point. We chose to focus on a large scale (of 1°) in this analysis because of few reasons. The link

between RR and IWP is not a simple link. Clouds' IWP change along their lifetime, and so is the rain production. This link is related to cloud type and environmental conditions as well. The satellite measure different clouds and at different instances along their lifetimes. So in order to reduce the complexity and to have a robust link between IWP and RR one has to look on large scales that represent cloud systems (10s to 100s of km) and not single clouds. TRMM-TMPA provides rain rate estimations in relatively high spatial resolution of 0.25°, to ensure that the grid box is larger than the typical footprint size for PMW precipitation estimates (~20 km). The MODIS Level-3 gridded product in 1° resolution is the most appropriate dataset for our research. It is more efficient for the study of global statistics, therefore we averaged also the TRMM-TMPA product to 1° resolution.

Following this comment, we edited the Data and Methods section:

*"The MODIS Level-3 gridded product (in 1° resolution) is suitable for global statistics and therefore for our research.. ... In order to match the resolutions of the two datasets (MODIS and TRMM-TMPA) we averaged the RR data into 1° pixels."*

4. Page 5, line 5: These rain rates should be described as "retrieved" or "estimated", not "measured" since strictly speaking only a gauge provides a direct measurement.

**Answer:** Thank you for this comment. We changed the text as suggested:

*"To obtain simultaneous cloud and rain estimations, we developed an Aqua time equivalent precipitation database by sampling the estimated rain rate data that were closest to the Aqua's passing time."*

5. Page 5, lines 7-8: The 3B42 rainfall rates represent an average over the entire 3-hour period rather than a value at a particular point in time; therefore, interpolation is probably not recommended; rather, the Aqua overpass should be matched with whichever 3-hour window of 3B42 is coincident with it.

**Answer:** Our main motivation was to create an Aqua-time (i.e. 13:30 local time) product of RR. For this reason, in each pixel we used 2 windows of 3-hr and a weighted average to calculate RR. In addition, we created an Aqua-time product without using an average

but taking the relevant window only. We calculated the β exponent and its $R^2$ using the two products. We detected no critical differences in the spatial distribution of $R^2$ (Fig. 4 a,c) or in the β values (Fig. 4 b,d). Histograms of $R^2$ and β values calculated using these two products are presented in the answer to comment 1 (see Fig. 3 above). We decided to present the weighted averaged product in this study.

[Figure]

Figure 4: Near-global distribution maps of (a, c) $R^2$ and (b, d) β exponent for June–July–August 2007 and for averaged (a, b) and no-averaged (c, d) Aqua time databases using the TRMM-TMPA 3B42 precipitation (PMW+IR) product.

6. Page 5, line 9: 3B42 provides estimates of rain rate, not rain amount.

**Answer:** Thank you for this comment. We changed in the text 'amount' to 'rate'.

*"The LWP or IWP are expected to correlate with rain rate as they represent the condensate mass of the cloud, which is the source of rain."*

7. Page 5, lines 19-27: It might be better to introduce this information after describing the calculation of the local means on line 5 of page 6 so that the purpose of the averaging is understood.

**Answer**: Thank you, we accepted this suggestion. This paragraph moved to page 6.

8. Page 5, lines 25-26: What is meant by the "trends" in line 25? Also, what was the basis determining what was "a good compromise between statistics [statistical significance?] and locality?"

**Answer:** Thank you for this comment. For clarifying this sentence we replaced the word 'trend' with 'spatial patterns'.

The frame of our research is a seasonal scale covering a quasi-global domain. We used two datasets with 1° spatial resolution. Our main focus is on moderate to heavy precipitating tropical convective systems and mid-latitudinal frontal systems with iced cloud-tops. The average size of such systems is between 10s and 100s of km. In order to fully cover such systems and avoid (as much as possible) the mixing of different cloud regimes we found the box size of 5 by 5 to be a good representative area. It compromises between statistics (large enough dataset for each analyzed box) and locality (representation of similar type of clouds). Below we present (Fig. 5) three examples for $\beta$ exponent maps as calculated for 3 different box sizes, 3x3, 5x5 and 9x9. We show limited coverage of $\beta$ values for the smallest cube of 3x3, spans mostly along the tropical belt and particularly the ITCZ. This limited coverage is due to small number of 3x3 boxes that contain sufficient data for analysis. The 5x5 box map shows larger spatial coverage including the entire domain except for the subtropical belts, as expected. Typical $\beta$ values here are in the range between 0.5 and 2. Using larger boxes (9x9) we get larger spatial coverage that partially covers subtropical areas as well. This is probably wrong as these regions are not characterized with moderate to strong raining systems. These areas show many irregular $\beta$ values lower than 0.5 (dark blue color in Fig. 5c) and small $R^2$ values (not shown). For this reasons we decided to focus on the 5x5 box size.

[Figure]

Figure 5: Near-global distribution maps of β exponent for June–July–August 2007 using 3 different box sizes: (a) 3x3, (b) 5x5, and (c) 9x9.

Following this comment we added a detailed description of the test performed before choosing the optimal analysis box size: *"Estimation of the link between IWP and RR requires a compromise between the size of the analyzed dataset (in the analysis box) and the locality of the results. On the one hand, more pixels yield better statistics, which is especially important for rain data that is highly variable. On the other hand, more pixels cover a larger area (less localized) that is more likely to contain different types of clouds (in different meteorological conditions). We tested different box sizes by conducting similar analysis changing only the box size (for boxes between 3x3 and 11x11), when the*

*basic pixel is 1° resolution). All those analyses showed similar spatial patterns, but the analysis that used very small boxes had poor total spatial coverage (due to smaller number of boxes that meet the required minimal number of IWP and RR pixels). On the other hand using too big boxes created false coverage in areas with no deep convective systems and outliers in the exponent values distributions (with small $R^2$ as well). Based on those tests we chose to analyze the data using 5 x 5 boxes (as moving window on the 1° pixels).”*

9. Page 6, Equation (3): It would appear that when simplifying Eq. 2, the α terms drop out and all that is left is Λβ. Where does θ come from here (and, consequently, in Eq. 4)?

**Answer**: Thank you for this comment. It is correct that when simplifying Eq.2 the α terms drop out and then we left with $\frac{IWP^\beta}{\overline{IWP}^\beta}$. When multiplying this term with $\left(\frac{1/\overline{IWP}}{1/IWP}\right)^\beta$ we get the term

$$\frac{\left(IWP/\overline{IWP}\right)^\beta}{\overline{IWP}^\beta/\overline{IWP}^\beta} = \frac{\left(IWP/\overline{IWP}\right)^\beta}{\left(IWP^\beta/\overline{IWP}^\beta\right)} = \frac{\left(IWP/\overline{IWP}\right)^\beta}{\left(IWP/\overline{IWP}\right)^\beta}$$

As introduced in the manuscript (page 6) $IWP/\overline{IWP} = \Lambda$ and $1/\Lambda^\beta = \Theta$, therefore the term in the line above equals $\Lambda^\beta\Theta$. When we calculate the log of the last term we get the same term as in Eq. 4 of the manuscript.

10. Page 7, lines 1-2: why were the data divided into 50 bins prior to creating the scatterplot, and what was the basis for binning? At first glance it would appear that this would make the results appear much less noisy than they really are.

**Answer:** We thank the reviewer for an important comment. Indeed when binning the data we might lose information and increase the apparent correlations. The main reason for doing so is when nonlinear relations are expected. In cases for which the natural variance is large (like in our data) if one wants to avoid a priory model to fit the data, then by

using the binned method the relation may emerge as the bins reduce the scatter. In order to minimize the effects of the bins on the overall trend we aimed in using as many as possible bins (keeping the averaging as local as possible) while gaining reduction of the scatter such that the nonlinear relations could be estimated in a significant way. We note that in order to keep the variance range within each bin as similar as possible the bins are of equal samples. We have tried arrange of bin numbers and eventually found out that 50 is large enough to capture the trend while reducing the overall scatter. The plot below demonstrates how the analysis using 50 bins enables capturing the different slopes of two 5x5 boxes, one in the tropical Indian Ocean (blue line) and second in the southern central Pacific (orange line). While most of the IWP data is smaller than 200 g m$^{-2}$ only 10 to 15 bins represent IWP data larger than 200 and enables good capturing of the β value

[Figure]

Figure 6 (Fig. 3d in the revised paper): Scatter plot of RR vs IWP for two 5x5 boxes, the blue dots and line for the Indian ocean (within box #2 in Fig. 7a below) and the orange dots and line for the southern Pacific Ocean (within box #4). This image is part of the revised Fig. 3 in the manuscript.

11. Page 9, line 8: this relationship is nearly linear ($\beta$=1); it would be informative to test for the statistical significance of $\beta \neq 1$.

**Answer:** We thank the reviewer for this important comment. Following this suggestion we add to the manuscript information on the differences between the tropics and the mid-latitudes $\beta$ distributions and mean values, and the statistical significance of the 'nearly' linear $\beta$ over the tropics.

We added the following text to the Results section in the revised manuscript:

*"The differences between the mean $\beta$ of the tropics and the mid-latitudes were statistically significant within a confidence level of 99% ($\alpha$=0.01 and p-value<0.01). We also found that the $\beta$ distribution over the tropical belt with 'nearly linear' mean $\beta$ of 1.01 were significantly different than a distribution with mean $\beta$ equals to 1 ($\alpha$=0.01 and p-value<0.01)."*

12. Page 9, lines 7-10 and elsewhere: it might be helpful to provide a plot or a few values showing how different the RR-IWP relationships are for typical ranges of IWP so that the significance of these differences (or similarities) between regions is clearer.

**Answer**: We thank the reviewer for this comment. As suggested we added to figure 3 additional subplot (3d) showing the relationship between RR and IWP for two different areas. We choose 2 boxes of 5° by 5° within the selected regions located in the tropical belt over the Indian Ocean (box #2) and southern Pacific Ocean (cube #4). Shown in Fig. 7d are two scatter plots of the RR vs. IWP for the two 5x5 boxes. Axes presented in linear scale in order to illustrate the differences between the nearly linear and the power law exponents ($\beta$) in a clearer way. The $\beta$ of the blue line is nearly linear and equals 1.05 ($R^2$=0.99) while the $\beta$ of the orange line is 1.62 ($R^2$=0.96).

[Figure]

Figure 7 (Fig. 3 in the revised paper): (a) Near-global distribution map of β values characterized by $R^2 > 0.8$ during June–July–August 2007. (b) Histograms (1–4) of the β values over four selected boxes (10 x 20 pixels), correspondingly numbered in (a), with the local mean β value, standard deviation and mean $R^2$ (see text in each histogram). (c) β histograms for the tropics (20° S – 30° N, blue) and the mid-latitude regions in both hemispheres (20° – 50° S and 30° – 50° N, red). (d) scatter plot of RR vs IWP for two 5x5 boxes, the blue dots and line for the Indian ocean (within box #2) and the orange dots and line for the southern Pacific Ocean (within box #4).

We changed the text in the Results section as follow:

*"Fig. 3d shows two scatter plots of the RR vs IWP for two 5x5 boxes, one located over the tropical Indian Ocean (within box #2) and the other one over the southern Pacific Ocean (within box #4). The axes presented in this figure are in linear scale in order to illustrate the differences between the power law exponents in a clearer way. The exponent of the blue line is nearly linear and equals 1.05 ($R^2$=0.99) while the exponent of the orange line is 1.62 ($R^2$=0.96)."*

We also add the relevant new information to the figure captions of Fig. 3:

*"(d) scatter plot of RR vs IWP for two 5x5 boxes, the blue dots and line for the Indian ocean (within box #2) and the orange dots and line for the southern Pacific Ocean (within box #4)."*

**Technical Corrections:**

1. Page 2, line 3: Trenberth appears to be the sole author of the paper cited here.

**Answer**: The citation was corrected to "Trenberth, 2011".

2. Page 3, lines 17-18: There is no entry for Thies et al. (2008) in the References.

**Answer**: Corrected. The reference was added to the list:

Page 16, lines 10-12: "Thies, B., Nau, T., and Bendix, J.: Precipitation process and rainfall intensity differentiation using Meteosat Second Generation Spinning Enhanced Visible and Infrared Imager data, J. Geophys. Res., 113, D23206, doi:10.1029/2008JD010464, 2008."

3. Page 13, lines 23-25: Hou et al. (2014) is not cited in the body of the manuscript.

Answer: Corrected. The reference was deleted from the list.

4. Page 14, lines 22-27: Lebsock et al. (2011) and Lin and Rossow (1997) are not cited in the body of the manuscript.

Answer: Corrected. The references were deleted from the list.

**Referee #2**

**General Comment:**

This manuscript reports on correlations between gridded precipitation and ice water path observations from TRMM and MODIS, respectively. The paper seeks to add to the literature concerning algorithms for retrieving convective precipitation intensity from visible and infrared radiance measurements through a somewhat different pathway that first involves retrieving IWP as opposed to directly correlating radiances to surface rainfall rate. The subject is appropriate for Atmospheric Measurement Techniques but there are fundamental with the manuscript that make it unsuitable for publication at this time. These include critical deficiencies in the approach, the omission of any direct evaluation of the technique, and a failure to acknowledge some fundamental circularity in the logic. Specifically, no justification is offered concerning the choice of datasets used in the study, there is no discussion of the impact of uncertainties in the IWP and RR estimates including the representativeness of visible and infrared-derive IWP in convective storms, and the physical significance of correlating rainfall and IWP at a resolution of 5x5 (i.e. much larger than the scale over which precipitation processes occur) is not discussed. In addition, little evidence is provided to support the claim that this technique might offer a more robust method for retrieving rainfall rate than those that have been developed in the past. Indeed, the very dataset the authors adopt to develop their statistical regressions already incorporates infrared estimates of rainfall intensity information that they purport to replace with this new algorithm. As a result of these concerns, which are elaborated below, I cannot recommend publication of this paper at this time.

**Answer:** We thank the reviewer for a detailed and deep review of our manuscript. We did a serious effort to respond to all the comments. We answer the general comment in a very detailed manner within the specific comments below. Please find our answers below in a point by point manner.

**Specific Comments:**

1. Despite the authors' claims concerning the novel nature of this technique, it is not clear how this work advances the state of rainfall intensity retrievals from visible and infrared radiances that already exist in the literature. While it is clear that ice water and rainfall intensity are physically connected through ice-phase precipitation processes, it is not clear that the specific approach presented here is any different from simply using these radiances directly to infer rainfall rate. By regressing separate IWP retrievals against rainfall rate, the method simply introduces the additional step of first estimating IWP from the raw radiances that can introduce its own uncertainties including assumed particle size, shape, vertical structure, sensitivity to large particles, and saturation at high IWP that complicate the subsequent relationship to rainfall rate. There is no mention of the influence of these sources of uncertainty in the manuscript. In fact, based on the fact that the authors use existing cloud products and do not perform any retrievals themselves, it is not clear they are aware of these issues or the fact that the measure of IWP used here is likely far from optimal for precipitating cloud scenes.

**Answer:** Following this comment we find it essential to clarify the objective of this manuscript. We want to show a quantitative link between surface RR and cloud macrophysical properties. In particular we attempt to provide additional way to estimate rain rates using remote-sensing retrievals of cloud properties. It is important to make clear we are not aiming to improve existing algorithms of RR that are based on raw radiances and nowadays are more oriented toward active microwave instruments, high resolution and real time estimations. Our focus is on the large scale, global and seasonal to climatological perspective. For this objective we took two well calibrated and tested cloud and rain products which are used, tested and improved for more than 15 years.

The MODIS-Aqua provides cloud optical and microphysical properties in a global coverage and daily resolution using passive sensors that measure radiation in the visible and near infrared (VIS-NIR) wavelengths. The retrieval of the IWP is based on a very clever algorithm that takes into account that it is a cloudy pixel in high certainty, define cloud-top phase and correct biases related to various factors such as the zenith angel, sun glint, partly cloudy pixels, surface type, wind speed, cloud shadows and atmospheric composition (Platnick et al., 2015). That is why we prefer to use the IWP product rather

than estimate it based on raw radiance. Assessment reports that examined different IWP products showed the highest agreement (for spatial distribution and magnitude) between the IWP retrieval of MODIS and the CloudSat's active profiling radar which retrieve the entire cloudy column (Eliasson et al., 2011). Several previous studies used in situ measurements and remote sensing estimations to observe and quantify the link between LWP and RR in liquid warm phase clouds (e.g. marine stratocumulus and cumulus; Comstock et al., 2004; Kubar et al., 2009; Chen et al., 2011). We are focusing here on total water path of mixed-phase deep convective clouds in order expand this field of research and quantify the relation of these clouds' IWP to RR.

The TRMM-TMPA 3B42 product has many advantages.  It is routinely producing rain rate estimations for almost 20 years (since 1998) with quasi-global coverage and relatively high resolution in space and time. It is based on multiple data sources including active precipitation radar (PR) and rain gauges. Its fundamental advantage is the calibration of the passive microwave (PMW) estimations with the TRMM active PR. In the current version (V7) the primary PMW-based instruments are the TMI, AMSR-E, SSMI, and SSMIS. For these instruments, observed brightness temperatures were translated to surface RR using an extensive library of hydrometeor profiles relating microwave brightness temperatures to surface precipitation rates based solely on the PR data (Huffman and Bolvin, 2017). In pixels that none of the TRMM's instruments or these HQ estimations are available during the 3-hour window, PMW estimations from the AMSU-B and MHS are used. Rain rate estimations using these last instruments are based on IWP–precipitation rate relations but they are also calibrated with the TMI-PR product. Only when no PMW observations are available it uses IR measurements. These calibrations together with evaluation of the TRMM-TMPA 3B42 product against surface measurements and surface radars make it a reliable rain product.

Considering the long period these products are being developed, evaluated and improved, together with their global gridded coverage with high resolution in space and time, make these two products very suitable to such research.

We changed the description of the 2 datasets in the Methods and Data section:

*"We used cloud properties retrieved by the MODIS algorithm (Platnick et al., 2003) on board the Aqua satellite. The MODIS algorithm uses the VNIR channels to retrieve the COT and droplet effective radius ($r_{eff}$) (Nakajima and King, 1990). The LWP and IWP are estimated as the product of the COT and $r_{eff}$ (Platnick et al., 2003). The MODIS algorithm of LWP and IWP takes into account only cloudy pixel that were retrieved with high certainty, defines their cloud-top phase and corrects biases related to various factors such as zenith angel, sun glint, partly cloudy pixels, surface type, and atmospheric composition (Platnick et al., 2015). This dataset provides daily data on a global coverage since 2002. We used level 3, 1° resolution daytime data collected around 13:30 local time. The MODIS Level-3 gridded product (in 1° resolution) is suitable for global statistics and therefore for our research.*

*Surface rain rate data was taken from the TRMM Multi-satellite Precipitation Analysis (TMPA) product 3B42 version 7 (Huffman et al., 2007). This product contains precipitation estimations based on various microwave instruments, geostationary IR, and surface rain gauges (Huffman et al., 2007). Fundamental advantage of this product is the inter-calibration of the PMW rain estimations with the TRMM active precipitation radar (PR). IR measurements are used only when PMW observations are not available (in our dataset it is less than 10% of the data, and it is located mainly over the southern mid-latitudes). Many previous studies evaluated the performance of the 3B42 V7 surface rain and found good agreement with surface measurements (e.g. Chen et al., 2013a, 2013b; Xue et al., 2013; Peng et al., 2014; Tang et al., 2015; Deo et al., 2017), especially for moderate to high rainfall rates. It is covers the tropics, subtropics and mid-latitudes, from 50°S to 50°N. This product is available from 1998, at 3-h and 0.25° resolution. In order to match the resolutions of the two datasets (MODIS and TRMM-TMPA) we averaged the RR data into 1° pixels. In our analysis we focused on June–July–August (JJA) and December–January–February (DJF) along 3 years (2006–2008)."*

2. More problematic is the fact that the precipitation dataset used in this analysis actually includes geostationary infrared radiance information in it. This guarantees a relationship between the TRMM precipitation estimates and associated cloud fields since similar

observations influence both retrievals. Other than a quick mention of this at the top of page 5, this circularity is not discussed at any length in the manuscript.

**Answer:** We thank the reviewer for this comment that helped us clarify this point. The TRMM-TMPA 3B42 precipitation product uses IR measurements only when no PMW observations are available. In order to evaluate the influence of the IR estimations on the precipitation data we used the *satellite precipitation source* provided by the TMPA 3B42 dataset to show the fraction of the IR out of the total data. The source probability histogram (Fig. 8a) shows that the IR estimations (i.e. source number 50, Huffman and Bolvin, 2017) are less than 10% of the total observations used in our dataset of JJA 2007. Moreover, we examined the spatial distribution of the IR measurements fraction (Fig. 8b) and found it is significant only in several spots over the southern hemisphere's mid-latitudes (latitudes 20° to 50°S). Along the tropical belt the IR fraction is also no more than 10% of the data (not shown). Note that this analysis was done using single 3-hr time frames (without averaging) in order to enable tracking of the observations source.

[Figure]

Figure 8: (a) Probability histogram of satellite precipitation source provided by the TRMM-TMPA 3B42 V7 in the Aqua time database for June–July–August 2007 (JJA 2007). Source index #50 is for the IR estimations. The additional sources within our dataset are the AMSU (index #1), TMI (#2), AMSR (#3), MHS (#6), AMSU&MHS avg. (#30), and conical estimates avg. (i.e. TMI, AMSR-E, SSMI, and SSMIS, #31) (b) Near-global distribution map of IR estimations fraction out of the total number of estimations for each pixel in the same database as in (a)

In addition, to validate our results we calculated the power law exponents (β) using only the High Quality (HQ) PMW precipitation data and compared it with the β values as calculated in the paper, using all observational sources (i.e. PMW+IR, the precipitation variable). Comparison of the 2 exponent maps (Fig. 9 a-b), shows high similarities in the β values and spatial distribution, except for areas that are contributed in a significant way by the IR data (such as the southern mid-latitudes). In this area the main differences are in the spatial coverage and values of the slope. $R^2$ histograms for the tropics and mid-latitudes (Fig. 9 c-d) show almost no difference in the $R^2$ distribution. The β histograms (Fig. 9 e-f) show almost no difference for the tropics and small decrease of the β in the mid-latitudes. This change is due to larger contribution of the IR data in the southern hemisphere, with relatively smaller exponents than for the HQ product. Such findings clarify the small and limited influence of the IR estimations on our β calculations.

[Figure]

Figure 9: (a) Near-global distribution map of $\beta$ value characterized by $R^2 > 0.8$ using the TRMM-TMPA 3B42 precipitation product (PMW+IR) during June–July–August (JJA) 2007. (b) similar to (a) but using the TRMM-TMPA 3B42 HQ precipitation product (PMW only). (c, e) probability histograms of $R^2$ and β-value for both precipitation products during JJA and for the tropics (20°S – 30°N). (d, f) similar to (c, e) but for mid-latitudes regimes of both hemispheres (20° – 50°S and 30° – 50°N)

We changed the Data and Methods section as follow:

*"...This product contains precipitation estimations based on various microwave instruments, geostationary IR, and surface rain gauges (Huffman et al., 2007). Fundamental advantage of this product is the inter-calibration of the PMW rain estimations with the TRMM active precipitation radar (PR). IR measurements are used only when PMW observations are not available (in our dataset it is less than 10% of the data, and it is located mainly over the southern mid-latitudes). Many previous studies evaluated the performance of the 3B42 V7 surface rain and found good agreement with surface measurements (e.g. Chen et al., 2013a, 2013b; Xue et al., 2013; Peng et al., 2014; Tang et al., 2015; Deo et al., 2017), especially for moderate to high rainfall rates."*

3. Another concern centers on the lack of error bars on any of the component datasets used in the analysis. What is the accuracy of the rainfall rates and ice water path estimates used? How does the accuracy of these products vary with rainfall intensity? The IWP estimates almost certainly 'saturate' in more convective cloud regimes and visible and infrared measurements are generally not sensitive to precipitation-sized particles from which rain actually forms. No attempt is made to compare the MODIS IWP estimates to those from collocated passive microwave observations provided by TRMM.

**Answer:** Thank you for this comment. We agree that a more detailed investigation of the variability and accuracy of the calculated exponent is required. We are aware of the uncertainty related to satellite observations of clouds and rain. It involves instruments' sensitivity, assumptions taken by the retrieval algorithm and errors in a priori information. Consider the difficulties and challenges to assess the joint-uncertainty when analyzing cloud and rain products we choose to describe the accuracy of our calculation by using the standard deviation. In order to do so we present first the standard error of RR (which is the dependent variable) for the specific case presented in Fig. 1 in the manuscript (5x5 box in the central Atlantic ITCZ). We show (Fig. 10c) that for the relatively low values of IWP (where we have most of the data) the standard errors are small. As IWP increases and less data is available the standard errors increases as well.

This graph describes well the high variability characteristic of surface RR in convective clouds which depends on the cloud type, cloud stage and environmental conditions.

[Figure]

Figure 10 (Fig. 1 in the revised paper): Mean maps of (a) ice water path (IWP, g m$^{-2}$) derived from MODIS-Aqua and (b) surface rain rate (RR, mm h$^{-1}$) derived from TRMM-TMPA for June–July–August 2007. (c) Scatter plot of RR against IWP using daily data for the entire season (92 days) and for a region of 5° × 5° (black square in (a)). The data is divided into 50 bins, each with 22 samples. The error bars represent one standard error for each bin (d) Similar to (c) but a scatter plot of $\log(\frac{RR}{\overline{RR}})$ against $\log(\frac{IWP}{\overline{IWP}})$ The red line represents the linear fit with slope $\beta = 1.07$ and $R^2 = 0.94$.

In addition we calculated for each pixel the standard error of the regression slope (SE) using the following equation

$$SE = \frac{\sqrt{\dfrac{\sum(y_i - \hat{y}_i)^2}{n - 2}}}{\sqrt{\sum(x_i - \bar{x})^2}}$$

We present in Fig. 11 the spatial distribution of SE. In the tropical belt which is characterized by stronger rain, SE values are the smallest (<0.05). On the other hand, over the mid-latitudes the SE values are larger (mostly in the range between 0.05 and 0.1). These findings are compatible with our standard deviation calculation showed in the paper (Figs. 3 & 4) for the tropics and mid-latitudes.

[Figure]

Figure 11: Near-global distribution maps of standard error of the regression slope for June–July–August 2007.

Following these analyses we added in the Results section the text relevant to Fig. 1c:

*"Figure 1c presents a scatter plot of the dependence of RR and IWP (non-zero 1° pixels), with error bars that represent one standard error for each bin. RR and IWP data is divided into 50 bins that contain equal numbers of samples."*

We also edited the figure caption of Fig. 1c: *"(c) Scatter plot of RR against IWP using daily data for the entire season (92 days) and for a region of 5° × 5° (black square in (a)). The data is divided into 50 bins, each with 22 samples. The error bars represent one standard error for each bin."*

4. While the sensitivity of the results to averaging-scale was 'tested' according to the authors, it is still not clear how the physical processes described earlier in the manuscript relate to IWP and rainfall rate estimates over a 5x5 grid box. How much of the 'signal' emerging from the regressions shown in Figure 1 is simply caused by the correlation between cloud fraction and rain fraction within these large boxes? Has any effort been made to normalize the results by cloud fraction and rain fraction to remove such effects? I do not believe that normalizing by the mean IWP and R over each grid box fully accomplishes this.

**Answer:** We thank the reviewer for this comment. We would like to clarify that the analyzed data is in 1° spatial resolution, and no averaging of these pixels were done. We used the larger boxes (5°x5°) for creation of a larger dataset that would support the statistics. Our main focus is on moderate to heavy precipitating tropical convective systems and mid-latitudinal frontal systems with iced cloud-tops. The average size of such systems is between 10s and 100s of km. In order to fully cover such systems and avoid (as much as possible) the mixing of different cloud regimes we found the box size of 5° by 5° to be  a good representative area. It compromises between statistics (large enough dataset for each analyzed box) and locality (representation of similar type of clouds). Below we present (Fig. 12) three examples for β maps as calculated for 3 different box sizes, 3x3, 5x5 and 9x9. We show limited coverage of β values for the smallest cube of 3x3, which spans along the tropical belt (Fig. 12a). This limited coverage is due to small number of 3x3 boxes that contain sufficient data for analysis. The 5x5 box map shows a much larger spatial coverage (Fig. 12b). Typical β values here are in the range between 0.5 and 2. Using larger boxes (9x9) we get larger spatial coverage that partially covers subtropical areas as well (Fig. 12c). This is probably wrong as these regions are not characterized with moderate to strong rain systems. These areas show many irregular β values that are lower than 0.5 (dark blue color in Fig. 12c) and small $R^2$ values (not shown). For this reasons we decided to focus on the 5x5 box size.

[Figure]

Figure 12: Near-global distribution maps of β exponent for June–July–August 2007 using 3 different box sizes: (a) 3x3, (b) 5x5, and (c) 9x9.

We changed the text in the Data and Method section:

*"Estimation of the link between IWP and RR requires a compromise between the size of the analyzed dataset (in the analysis box) and the locality of the results. On the one hand, more pixels yield better statistics, which is especially important for rain data that is highly variable. On the other hand, more pixels cover a larger area (less localized) that is more likely to contain different types of clouds (in different meteorological conditions). We tested different box sizes by conducting similar analysis changing only the box size*

*(for boxes between 3x3 and 11x11), when the basic pixel is 1° resolution). All those analyses showed similar spatial patterns, but the analysis that used very small boxes had poor total spatial coverage (due to smaller number of boxes that meet the required minimal number of IWP and RR pixels). On the other hand using too big boxes created false coverage in areas with no deep convective systems and outliers in the exponent values distributions (with small $R^2$ as well). Based on those tests we chose to analyze the data using 5 x 5 boxes (as moving window on the 1° pixels)."*

5. Each of these concerns brings up one additional overarching concern related to the choice of datasets adopted for this analysis: it is not at all obvious that the datasets adopted here are optimal for this study. Since TRMM carries the Visible and InfraRed Scanning radiometer (VIRS), ice water path estimates similar to those derived from MODIS observations can be compared against TRMM rainfall estimates from either the Precipitation Radar (PR) or Microwave Imager (TMI) directly avoiding the issues of collocating MODIS and TMPA. Collocating these two datasets from independent satellites merely introduces uncertainties owing to time/space collocation errors, restricts the time of day to 1:30 pm, introduces artificial correlations through the influence of IR measurements on the TMPA, requires analyzing larger grid boxes (see below), and precludes the use of passive microwave IWP estimates for independent validation. What's more, many of the visible/infrared rainfall retrieval techniques currently employed have been developed using these direct collocated datasets and are, therefore, likely more robust than the technique proposed here.

**Answer:** We would like to thank the reviewer for this comment. The selection of these two datasets of MODIS and TRMM support in a very good way the purpose of this study. We decided to focus on a large scale (climatological global scale) that represent well the link between IWP and RR for cloud systems. For that target we found the MODIS L3 gridded product and the TRMM-TMPA 3B42 to be very suitable datasets. MODIS cloud products are considered as well tested and validated products, with many advantages. Although the IWP calculation is based on two parameters (COT, $r_e$) that are retrieved simultaneously, its algorithms use 6 or more different channels. It also requires the cloud

mask, cloud top pressure, land and snow/ice spectral albedos and external ancillary data such as wind speed, temperature and water vapor from NCEP GDAS. In the current version the IR cloud phase retrieval went through extensive testing and evaluation using the CALIPSO/CALIOP products. Considering almost 20 years of investigation, evaluation and improvements, the cloud product algorithms include the cloud ice model, multilayer detection, clear sky restoral filtering, cloud phase, view angle biases, and the impacts of non- plane-parallel clouds (Platnick et al., 2015, and references within). These datasets also support one of our objectives, to compare this link between IWP and RR for the tropical belt and the mid-latitudes. Using only the collocated instruments onboard the TRMM is not appropriate for that, considering the TRMM spatial coverage (38°N to 38°S). Our main objective is to perform large-scale rain estimations as simple and as robust as possible using MODIS cloud optical products. Available for almost 20 years in a global daily resolution, such estimations will be very useful in the large-scale seasonal manner. More over taking into account the similarity of the MODIS product to the older AVHRR product or the current future mission of the Suomi NPP-VIIRS that continuing the MODIS such parameterization can be very useful also for long-term analyses and climate models.

6. That leads me to my final point – no independent evaluation or comparisons to other techniques is provided to support the claims concerning the benefits of this new approach. Evaluation against independent rainfall estimates is required to assuage fears concerning the flaws outlined above.

**Answer**: Thank you for raising this important issue. Remote-sensed global datasets of rain suffer from many types of uncertainties, some of them are due to the nature of surface rain (intermittent and local phenomenon). Therefore it is a problem to validate our method since there is no ground truth oceanic dataset of global rain rates. Nevertheless, as we understand the importance of evaluating our method we did it by using another global dataset of rain rates (PERSIANN-CCS; Hong et al., 2004). We are aware of the limitations of both datasets but we think it is still valuable to do this comparison in order to support our findings. The PERSIANN dataset provides quasi-global (60°S to 60°N) rain estimations in a high spatial (0.04°) and temporal (1-hourly)

resolution. It is primarily based on IR's brightness-temperatures measurements and its relationship to RR depends on cloud patch classification. Each cloud patch is classified based on texture, geometry (e.g. areal extent), dynamic evolution, and cloud top height. Classification of cloud patches and rain estimation are initially trained using active and passive microwave measurements from satellites and surface radars. We compared the $R^2$ and the power law exponent histograms of the PERSIANN analysis with the $R^2$ and β exponent of the original precipitation dataset used in our paper. In the PERSIANN analysis the calculated β with correlation coefficients higher than 0.8 were located mainly over the tropical belt (with almost no pixels over the mid-latitudes). Therefore the histograms that compare the 2 analyses (Fig. 13) represent the tropical belt alone (20°S-30°N). We found some clear similarities between the PERISANN and the TRMM-TMPA precipitation product especially the high correlation along the tropical belt which distributed similarly to the TRMM product and also the β value distribution. The mean β value and standard deviation of the PERSIANN product are 1.01±0.13 and for the TRMM-TMPA precipitation product are 1.01±0.18. We found that the PERSIANN exponents with $R^2>0.8$ has the same mean β and it also has the same range as the TRMM-TMPA product, spans mostly between 0.5 and 1.7. Nevertheless it has more values on the edges of this range therefore its standard deviation is larger. In the scope of our research we couldn't explain these small differences but showing two different independent rain products that produce β values in similar range and mean values, and also with high correlations strengthen our findings about our method and results. The fact that the primary input of the PERSIANN-CCS product is IR measurements cancel to our opinion the argument that the linear relationship we observed using the TRMM-TMPA is mainly due to the sensitivity of this product to IWP.

[Figure]

Figure 13: (a) $R^2$ and (b) β-value probability histograms for Aqua time database using the precipitation product (PMW+IR), and PERSIANN-CCS during JJA and for the tropics (20°S – 30°N).

Consider this results we add to the manuscript this evaluation with PERSIANN-CCS.

In the Methods section: *"In order to evaluate our findings on the link between RR and IWP we performed similar analysis using different and independent source of RR data. The PERSIANN-CCS (Hong et al., 2004) product provides quasi-global (60°S to 60°N) rain estimations in a high spatial (0.04°) and temporal (1-h) resolution. It is primarily based on IR's brightness-temperature measurements that are linked to RR based on cloud patch classification."*

In the Results section: *"For evaluating our results we compared the distributions of the β values of the TRMM-TMPA analysis (and the $R^2$) with the PERSIANN analysis results (not shown). We compared only the tropical belt results since it contains most of the matching pixels. We found for the pixels with $R^2>0.8$ similar mean β value (1.01) and similar range of β values (0.5-1.7) to the ones calculated using the TRMM dataset. Nevertheless the PERSIANN produced more extreme values in this range and therefore it shows larger standard deviation (0.13 vs 0.18 respectively)."*

In the Summary section: *"We evaluated our findings using the PERSIANN-CCS rain product which is different and independent of the TRMM-TMPA. High $R^2$ values calculated for the tropics, with similar mean $\beta$ value (1.01) and similar range of exponent values (0.5-1.7) strengthen our method and results."*

**References**

Behrangi, A., Lebsock, M., Wong, S. and Lambrigtsen, B.: On the quantification of oceanic rainfall using spaceborne sensors. *Journal of Geophysical Research: Atmospheres*, *117*(D20), 2012.

Cao, C., De Luccia, F.J., Xiong, X., Wolfe, R. and Weng, F.: Early on-orbit performance of the visible infrared imaging radiometer suite onboard the Suomi National Polar-Orbiting Partnership (S-NPP) satellite. *IEEE Transactions on Geoscience and Remote Sensing*, *52*(2), pp.1142-1156, 2014.

Chen, Y., Ebert, E.E., Walsh, K.J. and Davidson, N.E.: Evaluation of TMPA 3B42 daily precipitation estimates of tropical cyclone rainfall over Australia. *Journal of Geophysical Research: Atmospheres*, *118*(21), 2013.

Chen, Y., Ebert, E.E., Walsh, K.J. and Davidson, N.E.: Evaluation of TRMM 3B42 precipitation estimates of tropical cyclone rainfall using PACRAIN data. *Journal of Geophysical Research: Atmospheres*, *118*(5), pp.2184-2196, 2013b.

Chen, R., Li, Z., Kuligowski, R. J., Ferraro, R. and Weng, F.: A study of warm rain detection using A-Train satellite data, Geophys. Res. Lett., 38(4), L04804, doi:10.1029/2010GL046217, 2011.

Comstock, K. K., Wood, R., Yuter, S. E. and Bretherton, C. S.: Reflectivity and rain rate in and below drizzling stratocumulus, Q. J. R. Meteorol. Soc., 130(603), 2891–2918, doi:10.1256/qj.03.187, 2004.

Deo, A., Walsh, K.J. and Peltier, A.: Evaluation of TMPA 3B42 precipitation estimates during the passage of tropical cyclones over New Caledonia. *Theoretical and Applied Climatology*, *129*(3-4), pp.711-727, 2017.

Eliasson, S., Buehler, S.A., Milz, M., Eriksson, P. and John, V.O.: Assessing observed and modelled spatial distributions of ice water path using satellite data. *Atmospheric Chemistry and Physics*, *11*(1), pp.375-391, 2011.

Hong, Y., Hsu, K.L., Sorooshian, S. and Gao, X.: Precipitation estimation from remotely sensed imagery using an artificial neural network cloud classification system. *Journal of Applied Meteorology*, *43*(12), pp.1834-1853, 2004.

Huffman, G.J., Bolvin, D.T., Braithwaite, D., Hsu, K., Joyce, R., Kidd, C., Nelkin, E.J. and Xie, P.: Algorithm Theoretical Basis Document (ATBD), 2017.

Huffman, G. J., Bolvin, D. T., Nelkin, E. J., Wolff, D. B., Adler, R. F., Gu, G., Hong, Y., Bowman, K. P. and Stocker, E. F.: The TRMM Multisatellite Precipitation Analysis (TMPA): Quasi-Global, Multiyear, Combined-Sensor Precipitation Estimates at Fine Scales, J. Hydrometeorol., 8(1), 38–55, doi:10.1175/JHM560.1, 2007.

Kubar, T. L., Hartmann, D. L. and Wood, R.: Understanding the importance of microphysics and macrophysics for warm rain in marine low clouds. Part I: Satellite observations, J. Atmos. Sci., 66(10), 2953–2972, doi:10.1175/2009JAS3071.1, 2009.

Nakajima, T. and King, M. D.: Determination of the optical thickness and effective particle radius of clouds from reflected solar radiation measurements. Part I: Theory, J. Atmos. Sci., 47(15), 1878–1893, doi:10.1175/1520-0469(1990)047<1878:DOTOTA>2.0.CO;2, 1990.

Peng, B., Shi, J., Ni-Meister, W., Zhao, T. and Ji, D.: Evaluation of TRMM Multisatellite Precipitation Analysis (TMPA) products and their potential hydrological application at an arid and semiarid basin in China. *IEEE Journal of Selected Topics in Applied Earth Observations and Remote Sensing*, *7*(9), pp.3915-3930, 2014.

Platnick, S., King, M.D., Ackerman, S.A., Menzel, W.P., Baum, B.A., Riédi, J.C. and Frey, R.A.: The MODIS cloud products: Algorithms and examples from Terra. *IEEE Transactions on Geoscience and Remote Sensing*, *41*(2), pp.459-473, 2003.

Platnick, S., King, M.D., Meyer, K.G., Wind, G., Amarasinghe, N., Marchant, B., Arnold, G.T., Zhang, Z., Hubanks, P.A., Ridgway, B. and Riedi, J.: MODIS cloud optical properties: User guide for the Collection 6 Level-2 MOD06/MYD06 product and associated Level-3 Datasets. *Version 1.0*, 2015.

Tang, L., Tian, Y., Yan, F. and Habib, E.: An improved procedure for the validation of satellite-based precipitation estimates. *Atmospheric Research*, *163*, pp.61-73, 2015.

Xue, X., Hong, Y., Limaye, A.S., Gourley, J.J., Huffman, G.J., Khan, S.I., Dorji, C. and Chen, S.: Statistical and hydrological evaluation of TRMM-based Multi-satellite Precipitation Analysis over the Wangchu Basin of Bhutan: Are the latest satellite precipitation products 3B42V7 ready for use in ungauged basins?. *Journal of Hydrology*, *499*, pp.91-99, 2013.